# THE PROMISES AND PITFALLS OF LANGUAGE MODELS FOR STRUCTURED NUMERICAL DATA

## ABSTRACT

Autoregressive language models are increasingly capable of processing non-text data, such as images or audio. Are language models also a natural choice for numerical data, such as the 3D structure of molecules? In this work, we use quantum chemistry simulations as a case study in the challenges of applying language models to numerical data, building up a set of simple subproblems that can shed light on key design decisions. We show that language models lag behind domain-specific models on prediction tasks and provide evidence for and against different hypotheses that explain their failure. Many commonly identified pitfalls such as difficulty performing arithmetic operations and choice of discrete vocabulary fall short of explaining the behavior. In contrast, we show that capturing invariance properties exhibits a strong correlation with predictive performance. Finally, we provide a comparison of language models trained from scratch on numerical data with models pretrained on text. We show that text pretraining often provides a surprisingly limited advantage on prediction tasks, and can even hurt performance, despite prior work showing that text-pretraining can offer advantages.

## 1 INTRODUCTION

A popular goal in machine learning is building a generative model that operates on many data modalities simultaneously. Multiple modalities are useful because each modality can unlock a new source of information or control. For example, a web developer can provide a chat bot writing code with a sketch of a website, or a biologist can provide a desired function or structure to a model that generates protein sequences (Hayes et al., 2024). One modality that is particularly exciting for scientific discovery is 3D structures, which are intrinsically geometric objects, but which often co-occur with text descriptions or categorical features. Improving models of these structures facilitates easier drug discovery and materials design. A sequence model that successfully incorporates numerical data might be able to leverage the few-shot abilities of large language models when making predictions about numerical structures or act as alternatives to traditional simulations.

When we consider current state-of-the-art models for predicting the properties of 3D structures, however, language modeling approaches appear to lag far behind models that leverage domain-specific knowledge. In other domains, such as images (Wang et al., 2024), time-series (Ansari et al., 2024), and generative modeling of molecular structures (Flam-Shepherd and Aspuru-Guzik, 2023), autoregressive language models can compete with other state-of-the-art methods, but the same pattern does not seem to hold true for many predictive tasks on structured numerical data. To find some explanation, we need only look to similar trends observed on simple numerical tasks. Namely, despite many fundamental advances in modeling, it is still challenging to learn algorithms for simple numerical operations that generalize to new input sizes (Zhou et al., 2023). Several authors have speculated on the fundamental challenges language models may face in learning algorithms over numerical inputs, some of which might act as useful frameworks for considering language models on 3D structures and similar geometric objects.

In this paper, our goal is to explore the core challenges in applying language models to geometric objects by appealing to frameworks from prior work and running extensive experiments to assess their practical significance. To do so, we train thousands of language models to solve operations from linear algebra and simple physical modeling tasks. We show that some facets of language modeling,

such as causal masking, are not fundamental bottlenecks to current progress on numerical tasks, while properties that affect a model's invariance to a problem's symmetries seems to have a strong correlation with predictive performance. We also explore how tokenization and pretraining can affect the predictions and invariances of models, and show that finetuning state-of-the-art language models often leads to worse results than training new language models from scratch.

For reproducing our experiments, we release our code at `https://anonymous.4open.science/r/numerical-tokens-E44C/`

## 2  RELATED WORK

Learning simple arithmetic operations with language models is longstanding problem. Although addition and multiplication of integers or matrices are in the complexity class of algorithms that can be learned exactly by a transformer-based language model (Merrill and Sabharwal, 2023), they can still be difficult to learn in practice. Zhou et al. (2023) speculate that some numerical operations are challenging to learn because they lack a *simple* program that can be expressed by a causal transformer. For example, addition of two multi-digit integers can be challenging when digits are ordered from most significant to least significant, because causal attention has trouble building representation for a carry operation. McLeish et al. (2024) draw on similar observations to design an improved language model with expanded generalization abilities on addition and multiplication, by reversing number digits and providing special information about each digit's location. While these studies provide useful frameworks for reasoning about the challenges of language models, they do not study high-dimensional objects, which have much more practical relevance. We show that in many cases the same intuitions do not naturally extend to our settings and different challenges dominate.

Going beyond integer inputs, Charton (2021) shows that language models can learn basic operations from linear algebra like matrix addition, matrix multiplication, and eigenvalue computations. While we use some of Charton (2021)'s tasks in our study and draw inspiration from their numerical string encodings, our analysis differs in fundamental ways because of our underlying motivation to approximate calculations from quantum chemistry. To this end, we focus much more on the invariance properties of learned models, and introduce simple building blocks of physical models that are not studied by Charton (2021). In this way, our work is more closely related to the work of Flam-Shepherd and Aspuru-Guzik (2023), which shows that language models with standard training and simple tokenization methods can be used as strong generative models of 3D structures. However, like Alampara et al. (2024), we also find that predictive tasks display different dynamics compared to generative modeling and that language models are not competitive with best in class predictive methods.

Because we apply language models to numerical data and closely study their interactions with choices in tokenization, we also drawn on the work of Golkar et al. (2023), which introduces a continuous alternative to discrete tokenization in language models (xVal) on mixed categorical and numerical data. xVal models all numbers with a single token and uses a single weight and bias vector for inputs and outputs, instead of an embedding matrix for many numerical tokens. xVal is thus akin to transformers applied directly on continuous inputs or to graph neural network methods applied to numerical prediction problems. We find that xVal does lead to improvements in many settings and therefore provides valuable perspective on what facets of language modeling are most challenging when learning on geometric data.

As we are also interested in how text pretraining can act as a useful inductive bias, our work also intersects with work that applies text-pretrained models to zero-shot prediction on other modalities (Hegselmann et al., 2023; Gruver et al., 2024b). Although text pretraining holds the potential to help models learn general-purpose circuits over discrete sequences, we find that it is ultimately unhelpful in our tasks.

## 3  PRELIMINARIES

Before studying the empirical performance of language modeling methods, it's worth briefly considering the reasons why we might prefer discrete representations to continuous representations in general. We lay out a few of the trade-offs intrinsic in each approach below:

**Continuous sequences:** Each number is a floating point value, typically at the same precision as the weights of the neural network.

- **Pros:** (a) domain-specific properties (e.g. invariance/equivariance) have simple relationships with the model parameters. (b) order information is preserved in the input and in loss functions. (c) it is not necessary to learn an embedding matrix, or associated linear layers, which might be very large.

- **Cons:** (a) information contained in the scale of the numbers can be destroyed by normalization used to improve numerical stability. (b) modeling numbers of radically different scales can lead to numerical instability in the input. Transforming with $\log$ and $\exp$ can stabilize the input but have poor gradient behavior. (c) multimodality (mixed categorical and continuous variables) in the output space can be hard to represent.

**Discrete sequences:** Each number is converted to a string and then to a sequence of tokens, i.e. ["1", ".", "5", "6"], and corresponding integers.

- **Pros:** (a) distributions on sequences are densities over numbers without strong distributional assumptions or complicated losses. (b) input numbers do not need to be normalized. Numbers can in principle be large or small without causing fundamental problems, though length generalization is not guaranteed.

- **Cons:** (a) learning basic operations on numbers might require many samples because of a large vocabulary and complicated algorithms for operating on strings. (b) hallucination of non-number outputs.

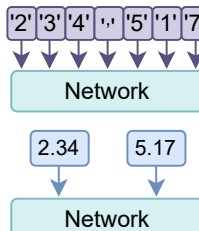

Figure 1: Discrete inputs mirror continuous inputs but have different costs and benefits.

As we can see, there are actually very reasonable explanations for supporting either approach, and working towards a sequence model over numerical tokens is not fundamentally misguided.

# 4 TARGET PROBLEMS AND BUILDING BLOCKS

**Target problems** As stated in the introduction, our goal is improving a practical application of modeling numerical data with language models. Here we focus on modeling of molecular structure for two primary reasons: 1) molecular structures tend to be intrinsically multi-modal because atom positions are numerical and atom identities (e.g., N or C) are categorical. In other works, this modality is often solved with complex diffusion/flow matching models (Campbell et al., 2024; Miller et al., 2024), but autoregressive sequence models offer a simple alternative. 2) molecular structures for crystals and organic molecules are often relatively small even for important problems, allowing us to study direct methods of tokenizing individual numbers without compressing chunks of the input, as in vector quantization. To this end, we use QM9 (Ramakrishnan et al., 2014) as a test bed.

In the Table 1, we show how a basic language model architecture (Touvron et al., 2023a) trained from scratch on QM9 compares to popular and state-of-the-art methods for predicting the highest occupied molecular orbital (HOMO). It is easy to see that language models are an order of magnitude worse than competing models, and in the following sections we will try to articulate a few possible reasons for this large gap.

Table 1: Comparison of popular and state-of-the-art approaches for predicting HOMO on QM9.

| Method | HOMO ($\downarrow$) |
|---|---|
| LLaMA (from scratch) | 212 meV |
| Non-equivariant GNN | 71.4 meV |
| Equivariant GNN (Satorras et al., 2021) | 51.9 meV |
| JMP-L (Shoghi et al., 2023) | 8.8 meV |

**Building blocks**  Our goal in this study is to develop a better mechanistic understanding of the challenges of applying language models to numerical data, and therefore we also attempt to break these goal problems down into their constituent parts and see where language models run into trouble. Both of the datasets we use are derived from density functional theory (DFT), quantum mechanical calculations to approximate the electron density for relatively small sets of atoms. We include a brief primer on quantum chemistry in Appendix A. At their core, DFT calculations rely on a relatively simple set of key operations. These include

- Looking up scalar constants (e.g. charge of an electron) or the value of a basis function (e.g. spherical harmonic) at a point.

- Matrix addition and multiplication (e.g. to compute pairwise distances between coordinates).

- Iterative procedures in linear algebra and solving differential equations (e.g. calculating eigenvalues or fixed point iteration on the Schrodinger equation).

This characterization is rough, but it gives a sense of what types of functions language models must be able to learn if they were implementing pre-existing theories of physics in order match the outputs of simulations. Of course, one of the strengths of neural networks is their ability to approximate expensive procedures with a fixed computation budget (one forward pass during training), so we do not necessarily expect language models to recapitulate existing tools from mathematics and physics. We can, however, use existing methods for approximating physical observations as a way of debugging current limitations of language models, if not to perfectly understand their internal mechanisms. If language models struggle to learn matrix multiplication but not matrix addition, for example, we can speculate that the scalar multiplication of many operands might be a roadblock, and we can work on this limitation directly.

**Theoretical limitions**  In the a single forward pass of a neural network, there are fundamental limits on both (1) the number of serial operations and (2) the amount of memory for intermediate results. When the number of steps that must be performed serially exceeds the depth of the network, the network will not be able to learn the exact function. Therefore, one forward pass of a network with depth 24 will not be able to learn an iterative method that might require 30 steps for convergence. Notably, this limitation is not relevant for functions that can be parallelized, as with matrix multiplication, which is in $TC^0$ and therefore should be possible to learn with a single forward pass (Merrill and Sabharwal, 2023). Like serial computation, memory can also be a bottleneck, as with computing the matrix product $A^T A$ where $A$ is $n \times m$ with $n >> m$. The number of computations will be approximately $O(n^2)$ but the input size is approximately $O(n)$. When $n$ is greater than the depth of the network, there can be challenges in storing all intermediate computations in the network's activations.

**Practical challenges**  In practice, reasoning about what solutions tend to be learned by a particular architecture is often more important than fundamental constraints. Even when a function can be represented in the function class, the statistical nature of the problem and questions of approaching the solution via approximations can play a larger role. Aside from not being able to represent the exact algorithm, why would autoregressive language models be limited in solving these numerical problems? We formulate 5 hypotheses and in the following sections provide evidence for and against each of these hypotheses.

1. *Mixing conditional and unconditional modeling* (section 6): Here we study predictive tasks $p(y|x)$, but language models parameterize a joint likelihood $p(x, y)$. Modeling the joint makes language models flexible but also requires more capacity for $p(x)$ beyond just $p(y|x)$. When $p(x)$ is intrinsically challenging, learning it can detract from learning a simple $p(y|x)$.

2. *Causal masking* (section 7): Features in autoregressive models are unidirectional, which makes learning some numerical operations challenging. For example, when digits are passed from left to right into a language model, it is challenging to express addition of two numbers using a carry bit. Similarly, any scan-style operation will be order-dependent and more challenging to learn if subsequences need to be reversed, and if a function cannot be implemented with an ordered scan-style operation, it might be very challenging to learn at all.

3. *Lack of symmetries* (section 8): Structured numerical data often obey constraints that are easy to express analytically (e.g. invariance to rotations). Incorporating these constraints can make

learning more sample-efficient or improve generalization (Frey et al., 2023), but language models are typically unconstrained.

4. *Poor tokenization* (section 9): Tokenization can lead to strange artifacts in text-pretrained language models (Brown et al., 2020; Wallace et al., 2019) and hinder their application to numerical inputs (Gruver et al., 2024a). While language models trained on numerical data often explore multiple tokenization schemes (Charton, 2021; Golkar et al., 2023; Flam-Shepherd and Aspuru-Guzik, 2023), they are often presented as ablations rather than analyzed in their own right.

5. *Too little data or pretraining* (section 10): Data for some numerical tasks can be relatively limited or extremely noisy, making language models less likely to succeed compared to models with more domain-specific assumptions. Other works show that text pretraining can serve as a surrogate for domain-specific pretraining or inductive biases (Gruver et al., 2024b).

## 5 EXPERIMENTAL SETUP

To test our hypotheses, we train thousands of language models that vary in model architecture, model size, tokenization method, loss function and pretraining method.

**String-encoding and tokenization** To turn numbers into tokens, we convert all numbers to a fixed precision and then convert these numbers to variable length strings by ignoring any leading zeros. These strings are then tokenized using a vocabulary of all numbers up to certain chunk length, for example {"1", "2", ..., "998", "999"} for a chunk length of 3. We greedily select the largest subsequence from right to left. For negative numbers, each negative number is prepended with "-". These strategies are similar to P10, P1000, and FP15 in Charton (2021), but, in our case, we choose to drop the exponent term used by Charton in favor of variable length because our inputs do not contain many different orders of magnitude. In addition to standard tokenization with an embedding matrix, we also explore Abacus embeddings (McLeish et al., 2024) and xVal (Golkar et al., 2023), which are tokenization methods specifically designed for processing numbers.

**Models** We present results for both language models trained from scratch and frontier language models pre-trained on text. Pretrained models can reuse general computational circuits and features developed on the pretraining text data, but may not be as well suited for numerical data in the given format. When training models from scratch, we use the LLaMA-2 (Touvron et al., 2023b) architecture with between 4 and 8 layers and hidden size 512, which translates to between 20 million and 50 million parameters. We train models with learning rate 0.0001 or 0.0005 and a cosine schedule. When studying pretrained models, we use LLaMA3.1-8B (Dubey et al., 2024), and the default LLaMA-3 tokenization, which, on numerical inputs, is identical to our 3-digit chunking method. We fine-tune the LLaMA3.1 models using LoRA with rank 8 and alpha 32 for one epoch. To make predictions with the models, we draw 10 samples at temperature 1 and calculate the median at each dimension of the output.

**Datasets** Our datasets are chosen to represent building blocks of common functions on numerical data. They have varying degrees of difficulty, with some being computable exactly by transformers while others can only be approximated. We explore two categories of tasks:

- **Linear algebra**: Following (Charton, 2021), we create $n \times n$ matrices with $n \in [2, 10]$ and evaluate (a) matrix addition (b) matrix multiplication, and (c) calculating real eigenvalues. We train on matrices of mixed sizes, with a distribution of $n$ weighted $n$, so that we train on more large matrices. The input matrices have coefficients sampled uniformly from $[-10, 10]$, and resulting eigenvalues having a center distribution with standard deviation $\sigma = 10\sqrt{n/3}$.

  These tasks have significant variation in difficulty. While matrix sum and product are computable in theory by a language model, computing an eigenspectrum is not and is more intrinsically serial than sum and product, making it more challenging for transformers. In addition to testing on matrices drawn from the same distribution as the train data, we also include a special generalization setting (marked with '+') in which we train on $n \in [2, 10] \setminus \{8\}$ and evaluate on $n = 8$. While past research often tests generalization evaluating on problems strictly larger than the training problems (Zhou et al., 2023), we opt for an interpolative setting because it is less confounded by the inherent limitations of position embeddings and reflects other facets of generalization on numerical data.

- **3D structures**: Using the data from QM9 (Ramakrishnan et al., 2014), we evaluate on the highest occupied molecular orbital (HOMO) regression task. We also evaluate on a set of simpler functions on QM9 coordinates including (a) calculating a distance matrix on coordinates, and (b) calculating a simple potential energy over the atomic nuclei. For the potential energy task, we test on either pre-computed distances or directly on coordinates, which disentangles the challenge of computing distances internally within the neural network, a task which can involve storing an intractable number of intermediate variables.

  Alongside the linear algebra tasks, these problems encompass many of the fundamental operations of quantum chemistry. It might be difficult to approximate current computation methods without being able to express reasonable approximations to these simpler problems.

For linear algebra tasks, we use 500,000 training examples, and for 3D structures we use 100,000 examples. We use 400 fixed test points for all evaluations.

**Baseline methods**  Our first baseline is low-precision quantization of the floating point numbers used in the correct computation within the synthetically generated tasks. We know that transformers struggle with performing exact arithmetic, even for integers, therefore we should expect that arithmetic will at best be performed approximately within the transformer. This quantization baseline evaluates the impact of using a correct algorithm but with only limited precision. We use QPyTorch (Zhang et al., 2019) and allocate an equal number of bits to the exponent and mantissa. Our two quantization baselines use 16 and 20 total bits, and this sets a reasonable ceiling on model performance.

Our other baselines are equivariant graph neural networks (EGNNs) (Satorras et al., 2021), which learn functions that are equivariant to permutations, rotations, and translations. EGGNs are therefore particularly useful in understanding how symmetries affect performance on our tasks. Following the original EGNN experiments on QM9, we use networks with 7 layers and hidden dimensions of size 128. Training details are included in Appendix B.

## 6 CONDITIONAL VS UNCONDITIONAL MODELING

In standard language modeling and in the supervised finetuning of language models, the joint distribution of the data is modeled, enforced by minimizing the NLL $-\log p(x) = \sum_i -\log p(x_i|x_{<i})$. For many of the problems we consider on structured numerical data, there is an explicit input-output structure, and we are only interested in the conditional distribution $p(y|x)$ for e.g. numerical outputs computed from a point cloud. Posed as sequence modeling we could also state $p(y|x)$ as $p(x_{>i}|x_{\leq i})$. While learning the joint distribution also implies learning the conditional distribution in the abstract, high complexity and variance in $p(x)$ can mean that signal in $y$ gets drowned out in the unnecessary task of modeling $x$. In many cases, the entropy of the output $H(y|x)$ is much lower than $H(x)$, and thus the model prioritizes $x$. For example, learning the distribution of all rotations of a molecule might be much more complicated than just learning to distinguish high and low energy configurations. When learning jointly on $p(x_{>i}|x_{\leq i})$ and $p(x_{\leq i})$, the gradient signals of each term compete, leading to slower learning of $p(x_{>i}|x_{\leq i})$ than in models that are explicitly conditional. Even if $p(x_{\leq i})$ is modeled perfectly, the random variation in $x_{\leq i}$ introduces unecessary noise in the gradients, which slows down learning as we show in Appendix C.

Table 2: MAE ($\downarrow$) values for training with and without masking, both from scratch and fine-tuning.

| Type | w/ | w/o |
|---|---|---|
| Scratch | 0.168 | 0.154 |
| Finetune | 0.456 | 0.508 |

To test this effect, we train models, both from scratch and fine-tuning text pretrained models, on the energy (from coordinates) task with loss masking to optimize only $p(x_{>i}|x_{\leq i})$. The results are displayed in Table 2. While masking helps fine-tuning which has relatively few gradient steps (1 epoch), masking does not help when training from scratch (100 epochs).

## 7 CAUSAL MASKING

In addition to whether or not the input $x$ is featured in the loss or is masked out, the decoder-only autoregressive structure of the language model has an impact on which operations are easy to

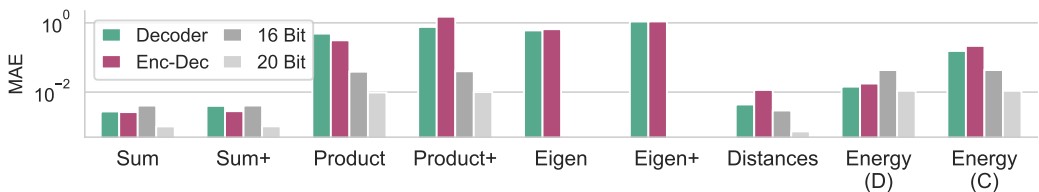

Figure 2: Encoder-decoder architectures have theoretical advantages over decoder-only architectures on our numerical tasks, but we find in practice the difference is minor. In our encoder-decoder models, layers are split equally between the encoder and decoder. A task name with '+' indicates a holdout of unseen matrix shape–a harder test of generalization. We include quantized numerical operations as baselines. `16 bit` refers a quantized operation with a 8 bit mantissa and 8 bit exponent. `20 bit` has a 10 bit mantissa and 10 bit exponent. We do not provide results for a quantized eigenvalue solver because PyTorch does not provide an easy mechanism constructing one.

express. For example, as identified by Zhou et al. (2023), the carry used in adding two numbers is a useful intermediary for the task, but if the numbers are ordered most significant digit first then its computation is nontrivial. In the structured numerical data context, an analogous challenge might arise when outputting scalar values which depend on aggregating information from input set data. For example, with an input $X = \{x_1, x_2, \ldots, x_n\}$, computing $y = \sum_{i,j} K(x_i, x_j)$. As a quadratic time operation that depends on all pairs of inputs, it might seem difficult for a causally constrained model to perform this computation within linear space allotted to the model.

To test this hypothesis, in Figure 2 (left) we compare the performance of a decoder-only model with loss masking to a encoder-decoder approach where only $y$ is modeled autoregressively and $X$ can be attended to bidirectionally by the encoder. We find that, contrary from intuitions from theory, encoder-decoder models do not perform significantly better than models with only causal masking. To enable an apples-to-apples comparison in these experiments, we use the same number of parameters in each architecture for each of three fixed parameter counts. In causal models, every layer is causal, whereas in encoder-decoder models, half the layers are in a bidirectional encoder and half the layers in a causal decoder. For tasks with a complex and high-dimensional output, it is possible that having a limited number of decoder layers could have a negative impact on the coherence of the output relative to a decoder-only architecture. This is one possible explanation for encoder-decoder architecture's significantly worse performance on calculating distances, where the output is a flattened upper triangular.

As a small additional experiment, we also explore McLeish et al. (2024)'s approach to tokenizing numbers, which involves reversing the the digits in number allowing for simpler algorithms implementing arithmetic operations Zhou et al. (2023). In addition to reversing the digit, a special embedding is added to identify each digit position within a number. Unlike the original paper, however, we use a plain decoder-only transformer model without parameter-sharing or skip connection to the input. In Table 3, we show that this intervention has a negligible or slightly negative effect overall. Although McLeish et al. (2024) designed their approach with large multi-digit numbers in mind, its surprising that there is no positive effect on learning operations that depend on addition and multiplication as a subroutine. Together, these two results (comparing architectures and input orderings) suggest that artifacts of causal masking are likely not the largest bottleneck to language model success on our tasks.

Table 3: Digit order has negligible effect on relative error. Geometric mean across tasks with standard errors.

| Method | MAE |
|---|---|
| Base | $0.237 \pm 0.12$ |
| Reversed | $0.309 \pm 0.14$ |

## 8 LACK OF HARD-CODED SYMMETRIES

Symmetries can be hard-coded into a model's architecture, but this process is not common in language modeling applications and can be challenging when operating on tokenized strings. In this section, we explore how language models learn invariances or equivariances on our numerical tasks and quantify

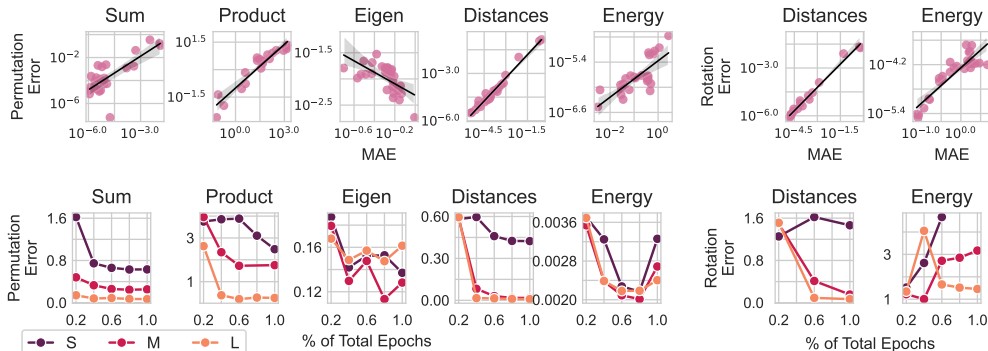

Figure 3: (**Top**) Degree of invariance (permutation or rotation error) strongly correlates with ability to fit the task (MAE) across several model sizes, tokenization methods, and training runs. Results are displayed with both axes log-scaled. For rotation invariance, we only study tasks on 3D structures. Surprisingly, when solving eigenvalues, predictive accuracy is inversely related to the permutation invariance of the model, which could be a result of a spurious correlation between the row-orderings and eigenspectra in the training dataset. Shading is a 95% confidence interval for the regression. (**Bottom**) Considering patterns of invariance over training, we see steady decreases in error in most cases, except on tasks where overfitting occurs. Larger models typically learn to be more invariant and become invariant more quickly. For linear algebra tasks, no form of augmentation is applied to the training data, while the tasks on 3D structures include both permutation and rotation augmentations.

how correlated learning symmetries is with predictive performance. In our linear algebra tasks, the primary symmetry is permutation equivariance. Permuting the inputs of addition and multiplication will lead to a permuted result, and the eigenvalues of a matrix are invariant to permutation. Our 3D structure tasks are equivariant to permutations, rotations, and translations, because all the tasks depend on only the relative positions of the atoms. We quantify invariance by calculating the predictions of the model for 10 examples transformed with random permutations or rotations. The invariance error is measured as the standard deviation per dimension normalized by the absolute value of the ground truth values and averaged over all dimensions. Following standard practice, we train our 3D structure models with rotation augmentations, and we also add permutation augmentations. For linear algebra tasks, we do not apply augmentations.

In Figure 3 (top), we show the correlation between predictive performance (relative error) and invariance to permutation or rotations. The points displayed are models that vary in size, architecture, and training hyperparameters. Across almost all tasks, there is a strong correlation, indicating that good models also tend to be invariant models. The notable exception is solving for eigenvalues, which displays the opposite trend, likely due to a spurious correlation between the matrix ordering and eigenvalue spectra in the training dataset. In many cases, the best models are able to approach perfect invariance, with invariance errors on the order of $10^{-6}$. Yet, even when models are nearly invariant, small changes in invariance appear to be correlated with improvements in performance. In Figure 3 (bottom), we show how invariance evolves during training and its relationship with model size. For most tasks, large models are able to quickly converge on invariant solutions, even when augmentations are not used.

To further explore the impact of equivariance, we compare decoder-only language models trained on digit tokens against GNNs with and without rotation equivariance in Table 4. GNN indicates permutation equivariance, while EGNN indicates permutation, translation, and rotation equivariance. The evaluation tasks are energy (from coordinates) and HOMO, both of which have permutation and rotation symmetry. In the results, invariance/equivariance again has a strong connection with predictive performance. Combined with the results above, we can conclude that invariance has a clear connection with performance on our numerical tasks. Surprisingly, language models can achieve high levels of invariance, but these high levels do not appear to saturate predictive performance.

Table 4: GNNs outperform LMs on the energy prediction task and benefit from equivariance.

| Method | MAE ($\downarrow$) |
|--------|--------------------|
| LM     | 0.209              |
| GNN    | 0.079              |
| EGNN   | 0.041              |

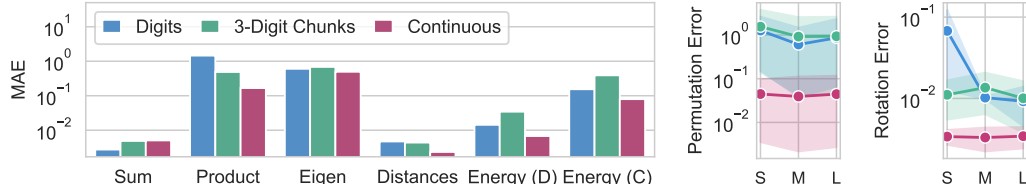

Figure 4: (**left**) We train causal transformers with different tokenization schemes and witness a significant advantage from learning a continuous prediction head. By contrast, differences between discrete tokenization schemes (digits vs. chunks) are inconsistent with multi-digit schemes performing better on some tasks and worse on others. (**right**) Using a continuous prediction head leads to higher invariance at smaller model sizes. For discrete methods, larger models are required to learn invariance. Numbers are the geometric mean over tasks, and shading denotes a 95% confidence interval.

## 9   TOKENIZATION

Beside architecture and training loss, tokenization is the other key design decision in constructing language models. When training on text, most language models employ tokenizers that compress commonly occurring sequences of bytes (e.g. byte-pair encoding). However, naively applying these same tokenization methods to numbers can lead to problems, because small changes to the value of the number can lead to large and hard to model changes in the tokenization of the number string (Gruver et al., 2024a). Character-level or n-gram tokens, therefore, are popular choices when modeling numbers, but while many papers employ these methods (Flam-Shepherd and Aspuru-Guzik, 2023; Zholus et al., 2024), there is little understanding of how tokenization affects the model's ability to learn basic numerical operations.

To test the effects of tokenization, we explore the empirical differences between tokenizing individual digits and tokenizing in 3-digit chunks. When using chunks, we always tokenize from right to left to maintain a consistent token meaning for strings of different lengths. In principle the primary trade-off between these approaches is between vocabulary size and sequence length, as chunked sequences are shorter but might require a larger training dataset to cover the space of $10^k$ tokens, for chunk size $k$, some of which might occur rarely. In addition to these two discrete approaches to processes numbers, we also run experiments with xVal (Golkar et al., 2023), which replaces discrete vocabularies and their associated embedding with a single linear projection that turns scalar inputs to vectors the same dimension as token embeddings and which projects final output layers. Instead of the cross-entropy loss, xVal uses an L2 loss on its continuous prediction. xVal is a useful counterpoint to purely discrete approaches because it sidesteps several key challenges of tokens, for example learning large vocabularies, long sequences, and potential challenges in learning symmetries.

In Figure 4, we show the effect of tokenization on predictive performance and symmetry learning. Overall we see that adopting a continuous approach leads to lower errors and more invariant predictors. By contrast, the difference between character-level (digit) and n-gram (3-digit chunk) schemes is inconsistent in terms of errors and nearly equivalent in terms of invariance. The latter result is surprising given our relatively large datasets, which provide reasonable coverage of the tokens in the vocabulary. The relationship between invariance and model size in xVal hints that maybe using any discrete representation incurs significant overhead to learn the appropriate structure, as xVal appears to acquire much higher levels of invariance for all model sizes.

In an attempt to understand the dominance of the continuous approach, we perform two additional ablations on the input and output of xVal by replacing them with their discrete counterpart, as shown in Figure 5. `Continuous Input` ablates the benefit of passing numbers directly into the model, without needing to parse inputs

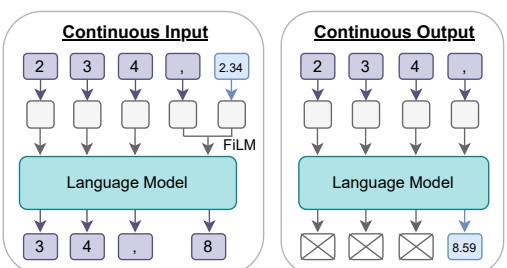

Figure 5: We include ablations on xVal to explore the effect of working with discrete versus continuous inputs and the corresponding loss functions.

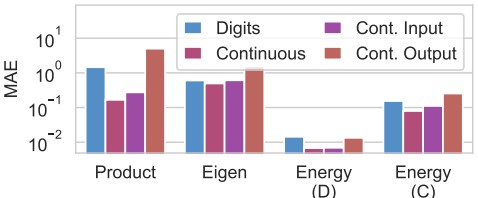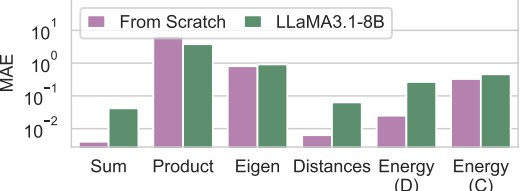

Figure 6: (**left**) To understand the performance of xVal, we perform an ablation the output and input with discrete tokens to understand if continuous inputs or continuous outputs (continuous loss) is the origin of improved performance. Both ablations hurt performance, but continuous inputs appear to be more helpful than continuous outputs. (**right**) We compare our small language models trained from scratch with large text-pretrained model finetuned with LoRA. Text-pretrained models perform worse on every tasks except matrix products, which might benefit from the pretrained model's additional capacity and ability to model high-dimensional outputs.

from a sequence of tokens, while `Continuous Output` ablates the benefit of using a continuous loss function, while still using discrete inputs. Figure 6 (left) shows the the result on the hardest numerical tasks, where the is worth investigating. The results indicate that neither design choice explains the strong performance of xVal in isolation, though continuous variants still outperform discrete approaches on 3D structure tasks.

## 10 INSUFFICIENT PRE-TRAINING

As we've seen so far, language models typically require large model sizes in order to capture invariances and make good predictions. For sufficiently large datasets, this allows language models to perform on-par with hand-crafted methods, but in other cases these extra parameters lead to poor generalization or slower convergence for fixed compute. The typical solution for this problem is extensive unsupervised pretraining, which can unlock the benefits of language modeling, while matching performance on narrow tasks. Prior work shows that text pretraining can serve this role in some cases. For example, Delétang et al. (2023) and Goldblum et al. (2023) show that text-pretrained models are general-purpose compression engines that can match domain-specific compression on non-text modalities.

To explore pre-trained models, we compare our small from-scratch models with LLaMA3.1-8B, a model two orders of magnitude larger. We fine-tune the LLaMA3.1 models for one epoch, which is 1-2 orders of magnitude fewer gradient steps than we take with the smaller models. As with models trained from scratch, we make predictions by drawing 10 samples and taking the median per dimension. Figure 6 (right) shows the resulting errors, for which pretrained models have worse performance in all but matrix multiplication. We posit two possible explanations for this discrepancy: (a) matrix product requires the most capacity to learn effectively (as was already evidenced in Figure 3) (b) matrix product has very high-dimensional outputs consisting of matrices containing large numbers, and text-pretraining is primarily helpful in modeling patterns in long sequences. If this were true, however, we might also expect some benefit on matrix addition and distance matrices.

## 11 DISCUSSION

In this work, we explored several explanations of the subpar performance of language models on 3D property prediction tasks. Through interventions like modifications of the architecture and loss function, we see that some of the explanations are not supported by the data, while others, such as the importance of invariances, hold up to scrutiny. We also showed that text pretraining is surprisingly unhelpful for learning good subroutines on our numerical tasks, despite its promise in other settings.

Our experiments suggest that language models converge to increasingly accurate and nearly invariant solutions when given sufficient model capacity and yet still have a large gap when compared to a method like xVal. One possibility for future work is to close this gap by extensively pretraining on synthetic data, which could be created by running cheap traditional methods (e.g. Hartree-Fock) or by distilling from a successful neural network potential evaluated on perturbed training data.

## 12 REPRODUCIBILITY

We include code to train, evaluate, and sample from language models in our code release. We include implementations for the exact architectures used in our experiments. The training and evaluation details for experiments we ran on each task were described by previous papers and again in our appendix.

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

# Appendix

## Table of Contents

## A  DENSITY FUNCTIONAL THEORY PRIMER

A core task in quantum chemistry is calculating the energy of a configuration of many atoms. Low energy configurations are stable and practically useful, for example in novel materials or the binding interface of therapeutic drugs. Atomic nuclei can be modeled as point charges

$$E_{\text{nuc}} = \sum_{i<j} q_i q_j / D_{ij}$$

where $D$ is the distance matrix between nuclei and $q$ is the charge of each nucleus. To model electrons, however, more complex methods are needed, for example, Hartree-Fock, which iteratively solves[1]

$$F(C)\, C = C\epsilon$$

where $F$ is the Fock matrix, $C$ are the orbital coefficients and $\epsilon$ is a diagonal matrix of molecular energies. At each step, $C$ and $\epsilon$ are obtained by solving a generalized eigenvalue problem using $F$ constructed from the last approximation of $C$, and, upon convergence, the electron energy is calculated as

$$E_{\text{elec}} = \text{Tr}(\epsilon) + \text{Tr}(C^{\dagger} H C)$$

where $H$ is the system's Hamiltonian (constructed using the position and charge of the atomic nuclei).

## B  GNN TRAINING DETAILS

We use a batch size of 96 and a learning rate of $0.001$ for 200 epochs on the HOMO prediction task and for 50 epochs on the synthetic energy prediction task from coordinates only. We use a learning rate of $0.0005$ for 100 epochs on the energy prediction task from distances. In all tasks, we use weight decay of $10^{-16}$ and a cosine decay on the learning rate. We do not use any normalization on the target function, and we add in an additional tanh activation function for stability.

## C  LEARNING SPEEDUP FROM LOSS MASKING

When learning $p(y|x)$, the training convergence can be substantially slowed down when including the $p(x)$ loss contribution.

Consider the loss for a single data point with a random label:

$$L = -y^{\top} \log \sigma(f_\theta(x))$$

---

[1] We show the Roothaan equations using an orthonormalised basis set

where $f(x)$ is the mapping to the log softmax of the logits of the model, $\sigma$ is the softmax function, and $y$ is the one-hot random label vector (among the $V$ classes).

The gradient is

$$\nabla_\theta L = y^\top [I - \mathbb{1}\sigma^\top]J$$

where $J$ is the Jacobian of the network outputs with respect to $\theta$. $\mathbb{E}[y] = \mathbb{1}/V$ giving an expectation of

$$\mathbb{E}[\nabla_\theta L] = (1/V)\mathbb{1}^\top[I - \mathbb{1}\sigma^\top]J.$$

The gradient is $0$ when the model predicts a uniform distribution $\sigma = \mathbb{1}/V$, and we will consider perturbations around this point.

From $\mathbb{E}[yy^\top] = I/V$ covariance is given by

$$\mathbb{E}[\nabla L \nabla L^\top] = (1/V)J^\top[I - \mathbb{1}\sigma^\top]^\top[I - \mathbb{1}\sigma^\top]J.$$

Letting $\sigma = \mathbb{1}/V$, the gradient norm is

$$\mathbb{E}[\|\nabla L\|^2] = (1/V)\mathrm{Tr}(PJJ^\top),$$

for $P = [I - \mathbb{1}\mathbb{1}^\top/V]$.

The convergence of SGD on convex problems can be written in terms of the expectation of the norm of the gradient. Over $T$ timesteps with learning rate $\eta$ and batch size $B$, the convergence can written (see e.g. Shalev-Shwartz and Ben-David (2014)) as

$$\frac{1}{T}\sum_{t=1}^{T}\mathbb{E}[\|\nabla L(\theta_t)\|^2] \leq 2\frac{L(\theta_0) - L(\theta_*)}{\eta T} + (\eta\sigma^2/B), \tag{1}$$

where $\sigma^2 = \sup_\theta \mathbb{E}[\|\nabla L(\theta, y) - \mathbb{E}[\nabla L(\theta, y)]\|^2]$ with expectations taken over the distribution of $y$. The convergence of SGD is limited by this noisy ball term $(\eta\sigma^2/B)$, and for a fixed learning rate cannot improve upon that limit as $T \to \infty$.

If $\mathbb{E}[\|\nabla L(\theta_*, y) - \mathbb{E}[\nabla L(\theta_*, y)]\|^2] = \mathbb{E}[\|\nabla L(\theta_*, y)\|^2] = (1/V)\mathrm{Tr}(PJJ^\top)$, then $\sigma^2 \geq (1/V)\mathrm{Tr}(PJJ^\top)$, therefore increasing the size of he noisy ball and loss value that SGD converges to.

For the $p(y, X)$ vs $p(y|X)$ scenario, $p(y, X)$ contains the additional random content of $X$ even if $y$ is a deterministic function of $X$. This random content when mixed in to the negative log likelihood objective increases the size of the noisy ball, slowing down convergence.

# D  HYPERPARAMETER SETTINGS

## D.1  FROM-SCRATCH MODELS

| Hyperparameter | Values |
| --- | --- |
| Model Size | {10M, 20M, 50M} |
| Model Dimension/Layers | {128/2, 512/4, 512/8 } |
| Learning Rate | {5e-4, 1e-4, 5e-5} |
| Tokenizer | {"1 Digit", "3 Digits", "Continuous"} |

Table 5: Hyperparameter values for from-scratch language model training runs.

| Hyperparameter | Values |
|---|---|
| Learning Rate | {5e-4, 1e-4, 5e-5} |
| LoRA Rank | {8, 16, 32} |
| Batch Size | {8, 16} |

Table 6: Hyperparameter values for fine-tuning language model training runs.

## D.2 FINE-TUNED MODELS

## E MAE NUMBERS WITH STANDARD ERRORS

Table 7 shows a full table of MAE values for each task and tokenization method, including standard errors calculated over 200 different examples from each task.

| Task | Tokenization | MAE | Standard Error |
|---|---|---|---|
| Distances | 1 Digit | 0.007583 | 0.001822 |
| Distances | 3 Digits | 0.007587 | 0.001873 |
| Distances | Continuous | 0.002345 | 0.000049 |
| Eigen | 1 Digit | 0.843819 | 0.051417 |
| Eigen | 3 Digits | 0.949008 | 0.056988 |
| Eigen | Continuous | 0.731731 | 0.044416 |
| Energy (C) | 1 Digit | 0.305043 | 0.016945 |
| Energy (C) | 3 Digits | 0.541269 | 0.029265 |
| Energy (C) | Continuous | 0.167922 | 0.009055 |
| Energy (D) | 1 Digit | 0.029822 | 0.005926 |
| Energy (D) | 3 Digits | 0.039592 | 0.006085 |
| Energy (D) | Continuous | 0.006789 | 0.001359 |
| Product | 1 Digit | 1.824334 | 0.131004 |
| Product | 3 Digits | 0.636723 | 0.055891 |
| Product | Continuous | 0.186717 | 0.006272 |
| Sum | 1 Digit | 0.003840 | 0.000149 |
| Sum | 3 Digits | 0.024948 | 0.006033 |
| Sum | Continuous | 0.005297 | 0.000159 |

Table 7: MAE values for different tasks and tokenization methods. Standard errors are calculated from 200 data points from each task.

