# OpenReview forum: "The Promises and Pitfalls of Language Models for Structured Numerical Data"
_ICLR.cc/2025/Conference — Submitted to ICLR 2025_

### Official Review · Reviewer_aFEH · 2024-11-04

**Soundness:** 3
**Presentation:** 3
**Contribution:** 3
**Rating:** 8
**Confidence:** 3

**Summary:**

This paper, "The Promises and Pitfalls of Language Models for Structured Numerical Data," explores the challenges and potential of applying autoregressive language models to structured numerical data, specifically within the domain of quantum chemistry. The authors conduct extensive experiments involving thousands of language models that vary across several dimensions: architecture, model size, tokenization strategy, loss function, and pretraining method. They demonstrate that language models often underperform compared to domain-specific models on predictive tasks involving structured numerical data, such as 3D molecular structures, and provide insights into potential reasons for these limitations. Key findings highlight that capturing invariance properties correlates strongly with predictive performance and that text pretraining provides limited benefit on these tasks. This study provides a comprehensive examination of the limitations of language models for numerical data and presents multiple areas where improvements could help bridge the performance gap.

**Strengths:**

I’m not an expert in this field, so my evaluation is based on my general experience in LLMs and deep learning.

- **Originality**: This paper’s originality lies in the hypotheses they propose and test regarding LLM training for structured numerical data, a relatively underexplored area, particularly in quantum chemistry. That said, I could be missing significant prior works, as I don’t have expertise in this specific area.

- **Quality**: This paper is a heroic effort to pack a lot of content into one submission. The authors demonstrate strong experimental rigor, testing thousands of models and covering multiple factors, including tokenization methods and model pretraining, which adds to the robustness of their findings.

- **Clarity**: Generally, the explanations are clear, and the figures effectively illustrate key findings. The authors also provide an accessible codebase for reproducibility.

- **Significance**: Given the rapid advancements in applying language models across diverse domains, this study addresses timely questions about LLM limitations on structured numerical data. These insights could help shape future research directions.

**Weaknesses:**

- **Overly Broad Scope**: The paper attempts to cover a vast amount of content in one submission, which brings two main challenges:

  1. **No Clear Takeaway Message**: With so much content, the reader might feel a bit overwhelmed, and after reading, may lack a clear takeaway message.

  2. **Lack of Depth in Analysis**: Some areas lack the depth they deserve. The broad scope results in a study that struggles to deeply focus on any single hypothesis. While the paper identifies key challenges (e.g., causal masking and tokenization), the current structure makes it difficult to evaluate each hypothesis thoroughly. Narrowing down to specific experiments would enhance the clarity and impact of each takeaway.

  My favorite part is Section 6 on conditional vs. unconditional modeling. I think this section could totally stand alone as a paper (assuming there’s little prior work thoroughly investigating mixing conditional and unconditional modeling), with more in-depth experiments and analysis. I find this idea quite straightforward, yet it’s one I hadn’t considered deeply before reading this submission. I’m curious if this phenomenon holds in a more general setting beyond the author’s structured numerical data context. While I’m not an expert in this field, this topic stands out to me as the most intriguing, and I’d appreciate deeper, more thorough experiments and analysis here.

  Overall, the current structure feels like a “bag of hypotheses”—probably correct but without a definitive answer for each. A paper that focuses on one hypothesis and runs every relevant experiment to add breadth and depth could deliver a much stronger message, rather than a few weaker ones.

**Questions:**

1. **On the Scope of the Paper**: Have the authors considered breaking the study into multiple papers? For instance, focusing on conditional vs. unconditional modeling in a separate work could allow for a more comprehensive analysis.

2. **Section 6 Expansion**: Could the authors provide more experimental results or analysis to investigate whether mixing conditional and unconditional modeling has general implications beyond their structured numerical data setting?

---

> ### Author Response · Authors · 2024-11-24
> **Rebuttal by authors**
>
> Thank you for your feedback and supportive words!
>
> With respect to your comment on **No Clear Takeaway Message**, we do think that there are some concrete takeaways from our experimental results, with practical implications on what researchers should focus on. For somebody hoping to apply language models to a numerical prediction task, such as predicting small molecule properties, we hope the paper inspires these conclusions:
> * If you can afford many epochs, use unconditional modeling over the input and target.
> * Apply a simple decoder-only model and don’t worry too much about the number of digits per token.
> * Encourage invariance through augmentations, and possibly consider endowing tokens with more numerical structure to encourage symmetry learning.
> * Do not use text-pretrained models. Prefer numerical pre-training data or forms of pretraining to aid learning invariances.
>
> We think that providing evidence that backs up these conclusions is a valuable contribution because it focuses the efforts of future researchers.
>
> We appreciate that you enjoyed the section on conditional vs unconditional modeling. We have not considered breaking the current paper up into multiple studies, and we feel like such a major modification is beyond the scope of acceptable revisions and would probably warrant a resubmission. Likewise, we don’t see a role for any experiments that extend beyond numerical data, as numerical datasets were our core focus in this submission.
>
> We hope that you will continue to support our paper for its originality and its experimental rigor. We think that our paper provides a valuable starting point for researchers seeking to understand language models applied to numerical data. Just as you were intrigued and inspired by the conditional modeling section, we believe many sections of the paper could inspire follow-up works, especially as the applications of language models continue to grow.

---

> > ### Comment · Reviewer_aFEH · 2024-11-27
> >
> > OK, what you said makes sense, and I agree these takeaways are interesting and valuable. I'd suggest highlighting the takeaways in the early sections of the paper, preferably in the introduction, so that the readers can have the claims and takeaways in mind before they go through the details. Do not force the readers to compile that information themselves. As this doesn't require major revision to the paper, I trust that the authors will improve readability in later versions and will raise my score from 6 to 8 given the extensive experiments and significant results.

---

### Official Review · Reviewer_kyCR · 2024-11-06

**Soundness:** 3
**Presentation:** 2
**Contribution:** 2
**Rating:** 5
**Confidence:** 3

**Summary:**

This paper explores several hypotheses of why language models have a hard time modelling linear algebra and the 3D structure of molecules (and by extension other numerical problems). The hypotheses include (1) conditional vs. unconditional modelling (conditional vs. joint distribution); (2) causal masking (encoder-decoder vs. decoder-only); (3) symmetry invariance; (4) tokenization schemes (discrete vs. continuous representations); and (5) natural language pre-training. The findings are that (1), (2) and (5) are not very important, whereas (3) and (4) are relevant (symmetry invariance is a strong predictor of final performance, and continuous embeddings outperform discrete embeddings).

**Strengths:**

I found the paper well-motivated and situated in the literature, and I appreciate the format of the paper (exploring a variety of hypotheses to generate insight into existing model behaviour). The number of experiments is sizeable and their setup seems well thought out and overall I don't have any concerns with the specific experimental setups.

**Weaknesses:**

Overall though, I found the narrative of this paper lacking. For someone who doesn't have a background in LLMs and structured numerical data I had a hard time imagining what sort of tasks exactly are being envisioned, i.e., what is the advantage of doing quantum chemistry with an LLM compared to using custom networks? (And is the goal here to study shortcomings of LLMs, or to try and find good models for structured numerical problems?) And given the experimental results, what do these experiments say about the path forward? Perhaps my lack of knowledge of this particular problem plays a role here, but the final recommendation of trying to "close [the gap between xVal and discrete inputs] by extensively pretraining on synthetic data" seems underwhelming. If the experiments don't suggest a clearer path forward, perhaps this particular experimental setup wasn't as insightful as it could have been?

I also think it would be useful to provide some small examples or illustrations of the problems being studied here (e.g., what a single HOMO task looks like).

**Questions:**

See above.

---

> ### Author Response · Authors · 2024-11-24
> **Rebuttal by authors (part 1)**
>
> Thank you for your review! We’re glad that you appreciated many aspects of the paper, and we hope to provide you with more narrative context and takeaways in what follows.
>
> We were inspired to write this paper by a flurry of work showing that language models can be used for molecular modeling [1,2,3]. These results suggested vanilla language models might be sufficient for good performance, others found that language models can fail for closely related tasks [4] or suggested alternative approaches to tokenization [5]. Motivated by a belief in the long-term potential of language models, we wanted to better understand when and why language models can come up short and how they might be improved. We’re seeking understanding, and we hope the paper performs some “mythbusting” by showing that many popular theories for understanding language models are limited or problematic.
>
> > *“what is the advantage of doing quantum chemistry with an LLM compared to using custom networks?”*
>
> LLMs are versatile and can handle multiple tasks—such as data analysis, hypothesis generation, and literature mining–within a single model. Custom networks are typically constructed by scientists who synthesize literature, distill hypotheses into model design, and tune hyperparameters. These steps could conceivably happen within a language model. Even without joint modeling of text and numerical data, however, there are many reasons we might prefer a “language model” approach in generative modeling of numerical data, where by “language model” we just mean learning a transformer-based autoregressive model over discrete inputs. We lay out some of these reasons in Section 3 in the paper, but we can also make a simple observation about trends in other areas of machine learning. In reinforcement learning [5], image modeling [6], and time series forecasting [7], language models are now frequently the state of the art. We wondered why the same trend does not appear to hold for all numerical tasks, and we don’t see any theoretical reason why language models could not conceivably reach top performance in these tasks as well, so we tried to identify key practical reasons for the performance gap.
>
> > *“is the goal here to study shortcomings of LLMs, or to try and find good models”*
>
> Our primary goal here was to understand why language models fall short on our tasks while they have succeeded elsewhere.
>
> > *“given the experimental results, what do these experiments say about the path forward?*
>
> For somebody hoping to apply language models to a numerical prediction task, such as predicting small molecule properties, we hope the paper inspires these conclusions:
> If you can afford many epochs, use unconditional modeling over the input and target.
> Apply a simple decoder-only model and don’t worry too much about the number of digits per token.
> Encourage invariance through augmentations, and possibly consider endowing tokens with more numerical structure to encourage symmetry learning.
> Do not use text-pretrained models. Prefer numerical pre-training data or forms of pretraining to aid learning invariances.
> We think that providing evidence that backs up these conclusions is a valuable contribution because it focuses the efforts of future researchers.
>
> > *“perhaps this particular experimental setup wasn't as insightful as it could have been?”*
>
> Our experimental setup is not perfect, but we think it does evoke the useful lessons above and potentially many more. We can see clear differences in the difficulty of each task and gain intuition, for example, challenges from quadratic memory usage when computing energies from coordinates. The connection between our experimental setup and tasks of practical significance should also be tangible. Operations from linear algebra and the other functions we study are key subroutines from quantum mechanics, as we touch on in more detail next.

---

> ### Author Response · Authors · 2024-11-24
> **Rebuttal by authors (part 2)**
>
> **Explaining HOMO**
>
> We agree that describing a HOMO calculation in detail might be helpful to readers that are less familiar with quantum chemistry. In fact, there is a clear connection between the linear algebra tasks and tasks like HOMO, which is worth highlighting.
>
> *Mathematical explanation*
>
> The Kohn-Sham equations in Density Functional Theory (DFT) are
> $$\left[ -\frac{1}{2} \nabla^2 + V_{\text{ext}}(\mathbf{r}) + V_{\text{Hartree}}(\mathbf{r}) + V_{\text{xc}}(\mathbf{r}) \right] \psi_i(\mathbf{r}) = \varepsilon_i \psi_i(\mathbf{r})$$
> with
> -  $-\frac{1}{2} \nabla^2$: Kinetic energy operator.
> -  $V_{\text{ext}}(\mathbf{r})$: External potential due to nuclei, defined as: $\begin{equation} V_{\text{ext}}(\mathbf{r}) = -\sum_{A} \frac{Z_A}{|\mathbf{r} - \mathbf{R}_A|}, \end{equation}$ where $\( Z_A \)$ is the charge of nucleus $\( A \)$ and $\( \mathbf{R}_A \)$ is its position.
> -  $V_{\text{Hartree}}(\mathbf{r})$: Hartree potential describing the classical electron-electron repulsion: $V_{\text{Hartree}}(\mathbf{r}) = \int \frac{\rho(\mathbf{r}')}{|\mathbf{r} - \mathbf{r}'|} d\mathbf{r}'$, where $\rho(\mathbf{r})$ is the electronic density.
> - $V_{\text{xc}}(\mathbf{r})$: Exchange-correlation potential, which incorporates quantum mechanical effects of exchange and correlation:  $V_{\text{xc}}(\mathbf{r}) = \frac{\delta E_{\text{xc}}[\rho]}{\delta \rho(\mathbf{r})}$, where $E_{\text{xc}}[\rho]$ is the exchange-correlation energy functional.
> -  $\psi_i(\mathbf{r})$: Kohn-Sham orbital, the single-particle wavefunction.
> -  $\varepsilon_i$: Eigenvalue corresponding to the $\( i \)$-th Kohn-Sham orbital.
>
> The Highest Occupied Molecular Orbital (HOMO) is the orbital corresponding to the largest eigenvalue $\(\varepsilon_i\)$ among all occupied orbitals:
>
> $$\varepsilon_{\text{HOMO}} = \max \{ \varepsilon_i \ | \ \psi_i(\mathbf{r}) \ \text{is occupied} \}.$$
>
> Taking a neutral carbon atom with six electrons and eigenvalues $\varepsilon_1 \leq \varepsilon_2 \leq \varepsilon_3…$, we can calculate the occupied orbitals as follows:
> - $\epsilon_1$: Occupied by 2 electrons (spin-up and spin-down)
> - $\epsilon_2$: Occupied by 2 electrons
> - ​$\epsilon_3$: Occupied by the last 2 electrons
> - Orbitals beyond ​$\epsilon_3$ remain unoccupied in the ground state.
>
> While this definition seems relatively complicated, we can see that it is simply the composition of a few simple operations, which include calculating distances and computing eigenvalues, all of which we test in our evaluations.
>
> **References**
>
> [1] Flam-Sheperd et al. (2023). “Language models can generate molecules, materials, and protein binding sites directly in three dimensions as xyz, cif, and pdb files”
>
> [2] Gruver et al. (2024). “Fine-tuned language models generate stable inorganic materials as text”
>
> [3] Sriram et al. (2024). “FlowLLM: Flow Matching for Material Generation with Large Language Models as Base Distributions”.
>
> [4] Alampara et al. (2024). “MatText: Do Language Models Need More than Text and Scale for Materials Modeling?”.
>
> [5] Chen et al. (2021). “Decision Transformer: Reinforcement Learning via Sequence Modeling”.
>
> [6] Yu et al. (2023). “Language Model Beats Diffusion – Tokenizer Is Key to Visual Generation”.
>
> [7] Ansari et al. (2024). “Chronos: Learning the Language of Time Series”

---

> ### Author Response · Authors · 2024-11-28
> **Kindly awaiting feedback**
>
> Dear Reviewer,
>
> We sincerely appreciate the time and effort you have dedicated to reviewing our submission and providing constructive feedback. As the rebuttal period approaches its conclusion, we want to ensure we address any remaining questions or concerns you might have.
>
> Has our response helped clarify the main takeaways of our paper, and can we provide any further discussion? Does our explanation of HOMO help contextualize the problems studied in the paper, and should we include any other helpful details?
>
> Thank you again for your thoughtful review and for helping us improve the paper.

---

### Official Review · Reviewer_JeY9 · 2024-11-09

**Soundness:** 2
**Presentation:** 3
**Contribution:** 3
**Rating:** 3
**Confidence:** 2

**Summary:**

The paper discusses the abilities of LLMs to model numerical data, including numbers in complicated data structures like a set of triplets of numbers used to encode 3D molecules.

First it trains a couple of Llama-like models (with and without pretraining) that fail to predict QM9 targets (compared to GNN baselines). Then it presents a few hypotheses and provides evidence for and against each of them.

**Strengths:**

* The problem setting in general is very interesting and relevant.
* Quite wide range of numerical tokenization methods were tested.
* The analysis of invariance and performance of the models is quite interesting. That's probably the most valuable results of this paper. It would be interesting to measure the degree of invariance for non-equivariant GNNs as well, to have a baseline for that particular equivariance metric.

**Weaknesses:**

1) The main weakness is that the language models tested are not sufficiently tuned for the tasks in question, which raises concerns whether the results presented in the paper relate to language models in general, or just to their suboptimal versions. The authors take a minified version of randomly initialized Llama 2 (<50M parameters), and a pretrained Llama 3.1 8B, and use one set of hyperparameters for each of them. I believe both models require large scale hyperparameter tuning, and maybe even separate tuning for each experiment!

LORA fine-tuning Llama 8B with just one epoch could be relevant for text-based tasks, but, to the best of my knowledge, there is no evidence that it is even close to optimal for a severe domain shift scenario like in the examples with linear algebra or quantum mechanics. Same is true for fine-tuning Llama 3.1 for only one epoch with such a small dataset (compared to trillions of tokens used in the original Llama).

Another argument that makes me think these trainings are suboptimal is that there are parallel works where open-sourced pretrained models achieve good results on downstream molecular prediction tasks. The examples I found are based on 2D representations (not 3D), but are quite competitive. Please check https://openreview.net/forum?id=p5VDaa8aIY , Table 8 on page 17. The same table also shows the impact of hyperparameter tuning. The hyperparameters borrowed from similar SFT (supervised fine-tuning) tasks from the same paper perform significantly worse than the hyperparameters specifically tuned for the BBBP task.

2) Table 2 shows that the variance of the methods is huge. It hints that all other tables and figures should also report the variance, and it's quite possible that Figure 2 results will become unsignificant.

Minor:
* The Table on line 150 probably refers to MAE, which is not mentioned. The table should have a caption. A downward arrow next to HOMO column would help readability. Table 3 says "MAE" in the column title. Please make them consistent.
* The paper uses two benchmarks: linear algebra and QM9. But most of the experiments are performed only on QM9.

**Questions:**

1) Can you please explain in more details what kind of spurious correlation can explain the negative correlation between invariance and prediction error for eigenvalues?

2) Did you do any hyperparameter search for Llamas?

---

> ### Author Response · Authors · 2024-11-24
> **Rebuttal by authors (part 1)**
>
> Thank you for your feedback. We address your comments and questions below:
>
> - **Hyperparameter tuning**: When training different architectures from scratch on each task, we trained with three model sizes, three learning rates, and three (or more) tokenizers, totalling about 30 runs per setting. Given that we were using highly optimized training recipes–i.e architecture/initialization (LLaMA), optimizer (AdamW), and learning rate (cosine annealing)--we view 30 runs as quite a large hyperparameter sweep. The submission that you referenced appears to sweep around 70 settings in Appendix A.6.1, which makes the degree of hyperparameter tuning fairly similar. Notably, many prior works have less than 10 hyperparameter settings per ablation [1,2,3]. When fine-tuning models with LoRA, we also swept across three learning rates, three settings of the LoRA rank, and two batch sizes. We have added these hyperparameter details to the appendix of the submission for clarity. Considering our diversity of evaluation tasks and limited access to compute, we view our hyperparameter search as pretty exhaustive.
>
> - **Using LoRA on numerical data**: We adopted LoRA because full fine-tuning was intractable given our available hardware. While we agree that LoRA is more limited than full fine-tuning, we don’t see any reason to believe that applying LoRA to numerical data is problematic, as several prior works have used LoRA fine-tuning successfully [4,5], and others show that language models can make high-quality predictions on numerical data even without fine-tuning [6,7,8]. LoRA’s out-of-distribution performance is itself an open research question, with some finding that LoRA performs well on out-of-distribution data [9].
>
> - **Statistical significance**: There might be a misunderstanding about what we are trying to show in Table 2. We presented the results to show that reversing the digits in line with [3] does not have a significant effect on performance. Combined with our experiments on encoder-decoder models, this observation suggests that interactions between casual masking and the order of numbers is probably not a major determinant of language model performance. As an assurance of the statistical significance of our other results, we include the tabular data for Figure 4 (left) with standard errors over the data points within each task:
> | Task       | Tokenization  |   MAE  |   Standard Error |
> |:-----------|:----------|:-----------|:------------|
> | Distances  | 1 digit        | 0.00758 | 0.00182  |
> | Distances  | 3 digits       | 0.00758 | 0.00187  |
> | Distances  | Continuous      | 0.00234 | 4.89e-05 |
> | Eigen      | 1 digit          | 0.843   | 0.0514   |
> | Eigen      | 3 digits        | 0.949   | 0.0569   |
> | Eigen      | Continuous      | 0.731   | 0.0444   |
> | Energy (C) | 1 digit         | 0.305   | 0.0169   |
> | Energy (C) | 3 digits        | 0.541   | 0.0292   |
> | Energy (C) | Continuous     | 0.167   | 0.00905  |
> | Energy (D) | 1 digit        | 0.0298  | 0.00592  |
> | Energy (D) | 3 digits        | 0.0395  | 0.00608  |
> | Energy (D) | Continuous      | 0.00678 | 0.00135  |
> | Product    | 1 digit        | 1.82    | 0.131    |
> | Product    | 3 digits        | 0.636  | 0.0558   |
> | Product    | Continuous      | 0.186   | 0.00627  |
> | Sum        | 1 digit        | 0.00383 | 0.000148 |
> | Sum        | 3 digits        | 0.0249  | 0.00603  |
> | Sum        | Continuous      | 0.00529 | 0.000159 |
> We didn’t display these error bars in the original draft because they were challenging to interpret when combined with the log-scaled y axis. However, we have now added them to the appendix.
>
> - **Invariance of eigenvalue models**: To better understand why there is a negative relationship between invariance and MAE, it helps to separate the models (displayed points) by size. If we consider only the small models, the correlation coefficient is -0.6 with p-value 0.07. If we consider the medium and large models, the correlation coefficient is -0.06 with p-value 0.77. Given that the correlation is a property of the small models, we investigate these models and find that among them, there were several models with high error and relatively low invariance because of repetitive erroneous predictions, which have low invariance because they do not effectively model the relationship between inputs and outputs (are overly invariant to all changes). Therefore the effect was not due to spurious correlations per se, but rather excessive invariance. We’re unsure why excessive invariance only occurs when using small models on the eigenvalues task, but possibly just because eigenvalues is the most consistently challenging task.

---

> ### Author Response · Authors · 2024-11-24
> **Rebuttal by authors (part 2)**
>
> **Re: Additional comments**:
>
> - *"The Table on line 150 probably refers to MAE, which is not mentioned. The table should have a caption. A downward arrow next to HOMO column would help readability. Table 3 says "MAE" in the column title. Please make them consistent."*: Thank you for your suggestions. We have made these changes in our revision. To clarify, Table 3 is measuring the MAE in predictions when averaging (via the geometric mean) is taken over a wide variety of linear algebra and 3D structure tasks. Therefore, this is not the same as in Table 2.
>
> - *“Most experiments are performed on QM9”*: While several of the small tables focus on results for QM9, the linear algebra results are also quite extensive and featured in Figures 2, 3, 4, and 6.
>
> We greatly appreciate your clarifying questions and we think they have helped us improve the paper. However, we think several of your criticisms are grounded in a misunderstanding of our experiment setup. We hope you will consider raising your score or ask for additional clarifications if there are any other potential sources of confusion.
>
> [1] Charton (2021). Linear algebra with transformers.
>
> [2] Golkar et al. (2023). xVal: a continuous number encoding for large language models.
>
> [3] McLeish et al. (2024). Transformers Can Do Arithmetic with the Right Embeddings.
>
> [4] Li et al. (2024). MMSci: A Multimodal Multi-Discipline Dataset for PhD-Level Scientific Comprehension.
>
> [5] Gruver et al. (2024). Fine-tuned language models generate stable inorganic materials as text.
>
> [6] Mirchandani et al. (2023). Large Language Models as General Pattern Machines.
>
> [7] Requeima et al. (2024). LLM Processes: Numerical Predictive Distributions Conditioned on Natural Language.
>
> [8] Deletang et al. (2023). Language Modeling Is Compression.
>
> [9] Hajipour et al (2023). SimSCOOD: Systematic Analysis of Out-of-Distribution Generalization in Fine-tuned Source Code Models.

---

> > ### Author Response · Authors · 2024-11-28
> > **Kindly awaiting feedback**
> >
> > Dear Reviewer,
> >
> > We sincerely appreciate the time and effort you have dedicated to reviewing our submission and providing constructive feedback. As the rebuttal period approaches its conclusion, we want to ensure we address any remaining questions or concerns you might have.
> >
> > If there are any points you feel require further clarification or discussion, we would be more than happy to provide additional details or engage in further dialogue to assist in your review.

---

### Meta-Review · Area_Chair_rGRy · 2024-12-23

**Metareview:**

This paper systematically investigates the capabilities and limitations of language models when applied to numerical data, particularly in quantum chemistry. The key findings include: language models underperform domain-specific models, text pretraining provides limited benefit, and invariance properties strongly correlate with performance. While the paper demonstrates strong experimental rigor with thousands of model evaluations and addresses an important question about LLM limitations, several critical weaknesses make it unsuitable for acceptance. Reviewer JeY9 identified major concerns about insufficient model tuning and potential statistical insignificance of results. The hyperparameter optimization was relatively limited compared to state-of-the-art approaches, raising questions about whether the observed limitations reflect inherent properties of language models or simply suboptimal training. There are also issues with experimental validation - high variance in results suggests many findings may not be statistically significant, and some key experiments were only performed on a single dataset. While the authors attempted to address these concerns during rebuttal by providing more details about hyperparameter tuning and statistical analysis, the fundamental concerns about experimental rigor remain. Based on this I vote to reject this paper.

**Additional Comments On Reviewer Discussion:**

The reviewers raised several significant concerns during rebuttal that were not fully resolved. Reviewer JeY9 highlighted major issues with model tuning and statistical significance - while the authors provided more details about their hyperparameter search and statistical analysis, questions remain about whether 30 runs per setting is sufficient given the complexity of the models and tasks. Reviewer kyCR questioned the paper's narrative and takeaways - although the authors provided a clearer list of conclusions, some concerns about practical applicability remain. Reviewer aFEH, while supportive overall, noted that the paper tries to cover too much ground without sufficient depth in any single area. While the authors made a good faith effort to address these concerns, providing additional experimental details and clearer takeaways, the core issues around experimental rigor and statistical significance were not adequately resolved. Of particular concern is that parallel work has shown better results on similar tasks with more extensive hyperparameter tuning, suggesting the paper's conclusions about language model limitations may be premature. Although one reviewer was satisfied with the responses and increased their score, the fundamental methodological concerns raised by Reviewer JeY9 support a reject decision despite the paper's interesting insights and extensive experimentation.

---

### Decision · Program_Chairs · 2025-01-22

Reject